# Synthesis of Hierarchically Ordered Porous Silica Materials for CO$_2$ Capture: The Role of Pore Structure and Functionalized Amine

Xiaoqi Jin *, Jinlong Ge, Liyuan Zhang, Zhong Wu, Linlin Zhu and Mingwen Xiong

School of Materials and Chemical Engineering, Bengbu University, Bengbu 233030, China;
63111960@qq.com (J.G.); 414528533@qq.com (L.Z.); 1019309881@qq.com (Z.W.); 154006801@qq.com (L.Z.);
517591353@qq.com (M.X.)
* Correspondence: l3167526736@163.com

**Abstract:** Hierarchically ordered porous silica materials (HSMs) with a micro/mesoporous structure were successfully prepared with the sol-gel method using dextran, dextran/CTAB, and CTAB as templates. The obtained hierarchically structured silica was successfully modified with amine groups through post-grafting and one-pot methods. Their architectural features and texture parameters were characterized by XRD, N$_2$ adsorption–desorption isotherms, SEM, TEM, FTIR, and TGA techniques. The results demonstrated that the pore structure depended on the reaction temperature and the amount of CTAB added in the synthesis procedure. A series of porous silica with hierarchical pore structures possessed abundant micropores, ordered mesopores, and a tunable surface area and pore volume. After modification, the ordered structure of the hierarchical porous silica almost disappeared due to the presence of amine groups in the pore channel. Furthermore, to explore the effect of pore structures and amine groups on CO$_2$ adsorption performance, before and after amine modification of HSMs, adsorbents were evaluated regarding the capacity of collecting CO$_2$ for comparison. According to these results, the varying microporous content, pore size distribution, and density of the amine groups were important factors determining the capacity of CO$_2$ capture.

**Keywords:** hierarchically ordered porous silica; dextran template; amine group modification; CO$_2$ capture

## 1. Introduction

As is well known, carbon dioxide is the most important "greenhouse" gas and is derived from the combustion of power plants, metallurgy, cement, chemical production, and human activities. The emission of carbon dioxide had caused many negative effects, such as climate warming, melting glaciers, and a rise in sea level, which even threaten human existence. To solve these problems, the development of carbon dioxide capture and storage (CCS) technology, including aqueous amine absorption [1], solid adsorbents adsorption [2], membrane-based technologies [3], and cryogenics for reducing and preventing CO$_2$ emission into the atmosphere, have received more and more attention in scientific research. Among CCS technologies, solid adsorbents adsorption technology has become a focus of current research because it is economical and eco-friendly and provides high energy efficiency and convenient operation. In the past few years, several research groups have designed new solid adsorbents with various pore structures for application in CO$_2$ capture and storage, including porous carbon [4], silica [5,6], zeolite [7], metal oxide [8], metal organic frameworks (MOFs) [9], and covalent organic frameworks (COFs) [10]. The research discovered that the adsorption process could be controlled by tailoring the pore structure, and the adsorption efficiency and selectivity could be improved by modifying the surface properties of these pore materials.

Among the studied solid adsorbents, ordered mesoporous silica materials, such as MCM-41, MCM-48, SBA-15, MSU, and HMS have been widely investigated regarding CO$_2$

capture owing to their high surface area, ordered tunable pore structure, and accessible functionalization [11–15]. These mesoporous silica materials possess several different mesostructures and pore sizes in the range of 2 to 10 nm, such as one-dimensional (1D), two-dimensional (2D) hexagonal, three-dimensional (3D) hexagonal, and cubic mesostructures, which exhibit obvious discrepancies in their capacity to capture $CO_2$. V. Zeleňák et al. [16] synthesized three kinds of mesoporous silicas with different structures—2D MCM-41 with a small 3.3 nm pore size; 3D SBA-12 with a 3.8 nm pore size; and 2D SBA-15 with a large 7.1 nm pore size, all modified with 3-aminopropyl moietiesand investigated the carbon dioxide sorption over these materials. They found that the $CO_2$ adsorption capability for the three amine-modified silica molecular sieves was dependent on the amine group surface density and the porous structure. A larger pores size and a three-dimensional structure for amine-modified molecular sieves were beneficial for enhancing the $CO_2$ adsorption capacity because they favored the improvement of the surface amine density and decreased the blockage of product diffusion through the pores. Meanwhile, Zhang et al. [17] also discussed the influences of silica supports (2D SBA-15, 3D sponge-like structure TUD-1, and 3D HS-5 disordered structure) and amine additive on the performance of $CO_2$ capture sorption. This research demonstrated that silica supports with a 3D pore structure exhibited higher amine functionalization efficiency and thus better $CO_2$ sorption capacity than those with a 2D pore structure. Unfortunately, there is a certain degree of limitation of mesoporous silica application in $CO_2$ sorption because the relatively thin pore wall and amorphous hole of the mesoporous silica easily collapse under the conditions of high temperature or moisture. Therefore, microporous zeolite materials and MOFs, due to their high surface area and excellent hydrothermal stability and selectivity, have been evaluated in the field of gas adsorption, particularly $CO_2$ adsorption. They have exhibited a high $CO_2$ adsorption capacity and thermal stability [18,19]. However, the disadvantages of microporous materials as adsorbents are mainly high molecular diffusion resistance and significantly decreased adsorption capacity as the temperature increases due to their small pore size and temperature sensitivity.

To overcome the abovementioned problems, the development of hierarchically porous materials by combining micropores and mesopores achieved efficient mass transport through the ordered mesopores, high surface areas from micro/mesopores, as well as selective accessibility of various sizes of species. Hierarchically porous materials with high surface areas, high pore volume ratios, high accessibilities, interconnected pore structures, facile mass transport properties, and high storage capacities have been used in various applications, in which they have demonstrated their superiority as adsorbents and catalysts compared with simple porous materials [20,21]. Currently, various strategies have been explored to prepare hierarchically porous silica with micro/mesoporous structures, such as the integration of different types of porous materials and the template method. These meaningful works attempted to establish composite materials as $CO_2$ adsorbents, which combined M41S- or SBA-based mesoporous silica with the microporous zeolite types ZSM, BEA, and NaX, such as ZSM-12/MCM-48, ZSM-5/SBA-16, Beta/KIT-6 and NaX/MCM-41 composites [22–25]. The hierarchical materials obtained from the integration technique exhibited advanced properties of microporous and mesoporous materials. However, the complex synthetic steps and the existence of an undesirable two-phase structure are the main drawbacks. Currently, using different mixed templates is regarded as one of the most effective ways to build hierarchical porous silica materials due to the simple and direct process of synthesis. The mixed templates include mixed solutions of cationic/nonionic poly(ethylene glycol) surfactants, triblock copolymers and lyotropic mixtures of amphiphilic block copolymers of different lengths with hydrophilic linear poly(ethylene oxide) (PEO) chains, and nonionic/ionic liquids (ILs) [26–29]. However, considering green and sustainable chemistry, this process of synthesis increases the fabrication cost and energy consumption because of the use of large amounts of expensive and toxic organic templates. Therefore, it should be developed into a mixed template method using an affordable template to prepare hierarchical porous silica materials.

As a template, natural polymers possess a great potential to prepare porous materials because of the advantages of low toxicity and low cost; they include cellulose, chitosan, and DNA duplexes [30–32]. Among them, dextran is a polysaccharide that is well-known as a non-toxic and biodegradable linear macromolecule that can easily be chemically modified and produce various derivatives. In particular, dextran exhibits excellent water solubility with active hydroxyl groups on the surface of its molecular structure, which is favorable for forming interactions between the neighboring units of the surfactant and inorganic species through hydrogen bonds, van der Waals force, and electrostatic force. Dominic Walsh et al. successfully prepared ordered zeolite materials using dextran as a template and obtained sponge-like silica/NaY composites, with porosity at both the nano- and macroscale, derived from the sacrificial dextran [33]. Using cross-linked dextran gel as a template, Bartłomiej Gaweł et al. [34] obtained hierarchical γ-alumina monoliths with macro- and meso- porosity. Meanwhile, porous metal oxide nanoparticles, including titanium, tungsten, and gadolinium oxides, have been consecutively synthesized using dextran as a template [35–37]. Thus, it is reasonable to consider dextran as a template to synthesize porous silica materials, but it is seldom reported.

In this work, we report the synthesis of hierarchically ordered porous silica materials (HSMs) with micro/mesoporous structures through the sol-gel process using natural polymer dextran as a template. We investigated the effect of the reaction temperature and CTAB content on the resulting silica structure. Controlled experiments showed that the reaction temperature, the molar ratio of dextran, and CTAB played important roles in the porous structure of the products. Moreover, the influence of the adsorbent structure of the hierarchical porous silica and the changes in the material's surface that caused the grafting of amine groups on the $CO_2$ capture capacity were explored. Furthermore, the structural features and textural parameters of the resultant hierarchical porous silica materials were investigated in detail using various characterization techniques, such as X-ray diffraction (XRD), scanning electron microscopy (SEM), transmission electron microscopy (TEM), $N_2$ sorption–desorption isotherms, Fourier transform infrared spectroscopy (FTIR), and thermogravimetric analysis (TGA).

## 2. Experimental Section

### 2.1. Materials

Cetyltrimethylammonium bromide (CTAB, A. R.) and dextran (($C_6H_{10}O_5$)$_n$ Mw = 20,000, A. R.) were provided by Aladdin Co., LTD. (Shanghai, China). Tetraethylorthosilicate (TEOS, A. R.), ethyl alcohol ($C_2H_5OH$, A. R.), (3-aminopropyl) triethoxysilane ($H_2N(CH_2)_3Si(OC_2H_5)_3$, APTES, A. R.,), N-[3-(trimethoxysilane) propyl] ethylenediamine (($CH_3O)_3Si(CH_2)NH(CH_2)_2NH_2$, NTPEA, A. R.), and hydrochloric acid (HCl, 37 wt%) were supplied by Sinopharm Chemical Reagent Co., Ltd. (Beijing, China).

### 2.2. Characterization

SEM images were obtained using a Zeiss Sigma 300 electron microscope at an acceleration voltage of 15 kVa. TEM micrographs were collected from a JEOL JEM 2100F at 200 kV. The powder XRD measurements were recorded on a Rigaku SmartLab SE diffractometer using a Cu Kα radiation (λ = 0.154056 nm) source in 2θ ranges from 1.0° to 10.0° with a scanning speed of 1.0°/min at 35 kV and 20 mA. The FTIR spectra were measured on a Nicolet iS10 analyzer using a scanned area of 4000–400 $cm^{-1}$ and a 4 $cm^{-1}$ resolution. The surface area and porosity were determined from the nitrogen adsorption–desorption isotherm obtained (at 77 K) using a Micromeritics ASAP 2020 sorptionmeter. The surface area based on the $N_2$ isotherm data was analyzed by BET (Brunauer–Emment–Teller) and the surface area and pore volume derived from the micropore were calculated by the t-plot method, the obtained parameters were exhibited in Table 1. The pore size distribution was determined from the adsorption branch, according to the Barrett–Joyner–Halanda (BJH) method as well as the density functional theory (DFT) equilibrium model. The narrow micropore size distributions and the pore volume were calculated from $CO_2$ adsorption isotherms

at 273 K on the basis of the DFT method. Thermal gravimetric analysis (TGA) data were recorded with a Netzsch STA 2500 Regulus analyzer. The samples were heated from 50 to 800 °C at a rate of 10 °C/min under a 7 mL/min $N_2$ flow. The ordered hierarchical porous silica was pre-treated at 150 °C for 8 h under a helium atmosphere. The $CO_2$ adsorption performance of the samples at 273 K and 298 K were measured using a Micromeritics 2460, and the results are shown in Table 1. Prior to each adsorption experiment, the sample was degassed for 8 h at 150 °C to remove the guest molecules from the pores.

**Table 1.** The structure parameters of all related samples.

| Samples | $S_{BET}$ [a]/$S_{mi}$ [b] $(m^2/g)$ | $V_t$/$V_{mi}$ [b]/$V_{narrow\ micro}$ [c] $(cm^3/g)$ | Carbon Dioxide Uptake [d] 273/298 K (mmol/g) |
|---|---|---|---|
| HSMs(D)-30 | 269/218 | 0.13/0.09/0.003 | 1.10/0.65 |
| HSMs(D)-50 | 401/70 | 0.23/0.03/0.003 | 1.00/0.57 |
| HSMs(D)-80 | 406/175 | 0.19/0.08/0.004 | 1.22/0.74 |
| HSMs(DC)-2 | 601/172 | 0.28/0.07/0.006 | 1.26/0.70 |
| HSMs(DC)-0.5 | 801/751 | 0.42/0.37/0.033 | 1.33/0.74 |
| HSMs(DC)-0.15 | 1045/953 | 0.58/0.53/0.034 | 1.64/0.86 |
| HSMs(DC)-0 | 1052/972 | 0.68/0.61/0.017 | 1.34/0.74 |
| HSMs(DC)-0.15-APTES | 781/703 | 0.45/0.39/– | 1.98/– |
| HSMs(DC)-0.15-NTPEA | 603/521 | 0.31/0.27/– | 2.22/– |
| HSMs(DC)-0.15-$NH_2$ | 120/68 | 0.12/0.06/– | 1.38/– |

[a] The BET surface area was calculated in a relative pressure range $p/p_0$ = 0.05–0.3. [b] The surface area and pore volume were estimated using the t-plot method. [c] The narrow micropore volume was calculated for $CO_2$ adsorption isotherms at 273 K using the DFT method. [d] $CO_2$ uptake at 273/298 K and 1 bar.

*2.3. Preparation and Functionalization of Ordered Hierarchical Porous Silica Materials*

Ordered hierarchical porous silica materials with micro/mesoporous structures were prepared using dextran as a template via the sol-gel technique. First, an aqueous–alcoholic dextran solution was prepared by mixing 36 mL of distilled water, 15 mL of ethanol, and 2.3 of g dextran. Subsequently, 9 mL of TEOS and 54 μL of HCl aqueous solution (0.67 mol/L) were dissolved in 30 mL of ethanol, and then this mixture was further added to the above mixture dropwise at room temperature. The resulting mixture was stirred at the preset temperature for 6 h. After cooling to room temperature, the solid products were filtered and washed several times. The final products were dried in an oven overnight at 120 °C. The dextran template was removed by calcination at 550 °C for 6 h. The reaction temperature was adjusted to 30, 50, and 80 °C, respectively. The obtained series of ordered hierarchical porous silica materials samples were labeled HSMs(D)-x, where x represented the different reaction temperatures of 30, 50, and 80 °C.

Ordered hierarchical porous silica materials were prepared using the dextran and CTAB dual template via the sol-gel technique. An aqueous–alcoholic dextran/CTAB solution was prepared by mixing 36 mL of distilled water, 15 mL of ethanol, and a certain amount of dextran and CTAB. The remaining steps followed those used for HSMs(D). The obtained products were labeled HSMs(DC)-y, where y = n_dextran/n_CTAB (y = 2, 0.5, 0.15, 0, representing the molar ratio of dextran/CTAB.

The modification of HSMs(DC)-0.15 hierarchical porous silica by amine groups was carried out by post-grafting and one-pot methods, respectively. The details of the procedure of the post-grafting method are as follows: in ethanol using (3-aminopropyl) triethoxysilane (APTES) or N-[3-(trimethoxysilane) propyl] ethylenediamine (NTPEA) as the amine source, 0.5 g silica was dispersed in 30 mL of ethanol under ultrasound for about 5 min. Afterward, 3 mL of APTES or NTPEA was added dropwise to the suspension. The resulting mixture was refluxed at 80 °C for 6 h under magnetic stirring. After cooling, the obtained solid was separated by centrifugation and washed with ethanol. The modified products were dried at 60 °C overnight and denoted as HSMs(DC)-0.15-APTES and HSMs(DC)-0.15-NTPEA.

Following the typical one-step co-condensation method, an aqueous–alcoholic dextran/CTAB solution was prepared by mixing 36 mL of distilled water, 15 mL of ethanol,

and dextran/CTAB with a molar ratio of 0.15. After that, 10 mL of APTES was added drop-wise to the aforementioned mixture, and then 54 μL of HCl aqueous solution (0.67 mol/L) was dissolved in this mixture. The remaining steps were in accord with those followed for HSMs(DC)-0.15, and the obtained product was labeled HSMs(DC)-0.15-$NH_2$.

## 3. Results and Discussion

### 3.1. Structural and Textural Features

Figure 1 shows the XRD patterns of HSMs(D)-30, HSMs(DC)-0, HSMs(DC)-0.15, and HSMs(DC)-0.15-$NH_2$. As can be seen, Figure 1 presents a sharp diffraction peak, indicating the existence of an ordered mesoporous structure for the HSMs(D)-30 sample synthesized with dextran as a template. Figure 1 also shows that the intensity of diffraction peaks for HSMs(DC)-0 with CTAB as a template and HSMs(DC)-0.15 with dextran and CTAB as templates were relatively reduced compared with that of HSMs(D)-30. In addition, the $2\theta$ angle shifted from 1.46° to 2.0°, and the corresponding $d_{(100)}$ space value decreased from 6.16 to 4.42 nm, implying that the additive CTAB template affected the pore order degree. After modification with amine groups, as revealed in Figure 1, a notable reduction in peak $d_{(100)}$ intensity was observed for HSMs(DC)-0.15-APTES compared with that of HSMs(DC)-0.15, implying that the organic functional group modification of HSMs(DC)-0.15 affected the pore order structure. These observations illustrated that the ordered pore silica samples were successfully synthesized using the sol-gel method with dextran, dextran–CTAB, and CTAB as templates. The XRD patterns of the samples HSMs(D)-50 and HSMs(D)-80 also are shown in Figure S1 (Supporting Information). These results suggest that the ordered porous structure deteriorated with the increase in the reaction temperature during the synthesis process.

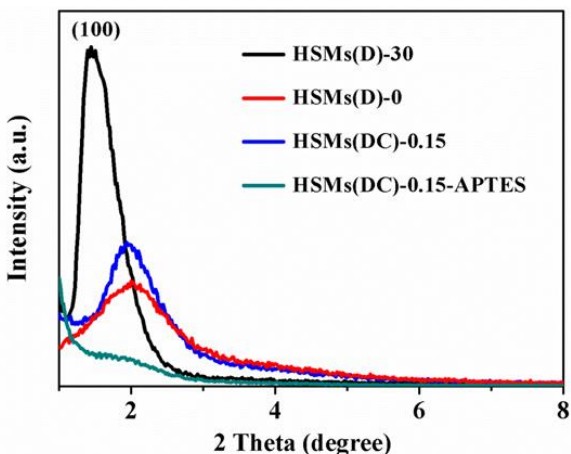

**Figure 1.** XRD patterns of HSMs(D)-x, HSMs(DC)-y, and samples after amine modification.

The morphologies and microstructures of the prepared samples were characterized by SEM and TEM, as shown in Figure 2. The SEM images for all samples show an irregular bulk construction. As the amount of CTAB increased, the larger porous silica began to be synthesized. The high-magnification SEM images of HSMs(DC)-0.15 are shown in Figure S2, the SEM images clearly display abundant mesopore and micropore structures with different pore sizes inside HSMs(DC)-0.15. Moreover, the rich hierarchical porous structure is directly observable in the TEM images shown in Figure 2(a1–c1). The high-magnification TEM images (Figure 2(a2–c2)) clearly demonstrate that HSMs(DC)-0.15 and HSMs(DC)-0 had a mesoporous structure and a micropore structure with irregular arrangement. The mesopores size was around 2.0–4.0 nm, and the size of the micropores was less than 2 nm, which was consistent with the following observation of the pore size distribution curves obtained from the $N_2$ adsorption isotherm and $CO_2$ adsorption isotherm (Figure 3b,d,e).

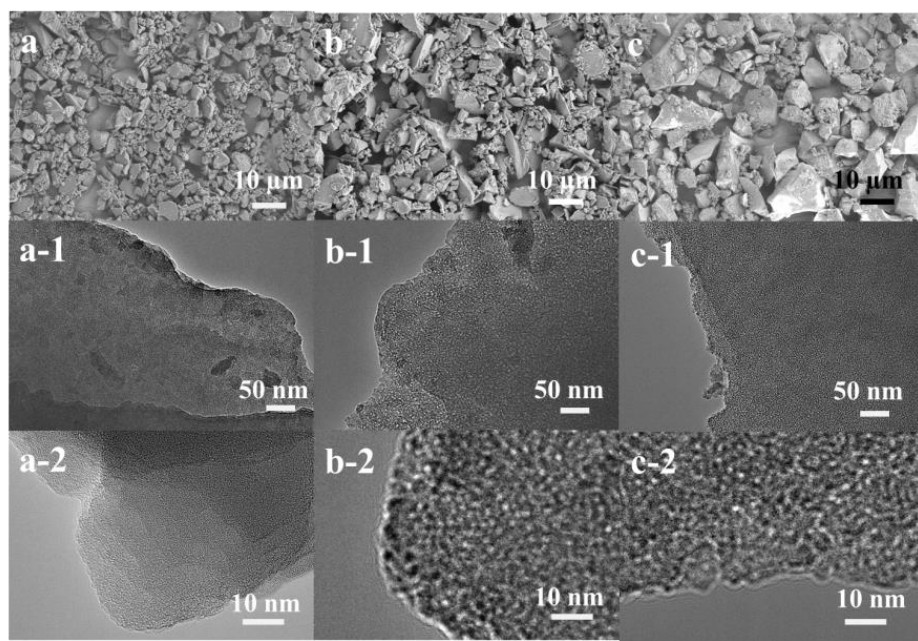

**Figure 2.** SEM images (**a**) HSMs(D)-30, (**b**) HSMs(DC)-0.15, and (**c**) HSMs(DC)-0. The corresponding TEM images of (**a1,a2**) HSMs(D)-30; (**b1,b2**) HSMs(DC)-0.15 and (**c1,c2**)HSMs(DC)-0.

The isotherms of $N_2$ adsorption–desorption and the distributions of pore size for the HSMs and amine-modified samples are shown in Figure 3. The pore size distributions for HSMs were analyzed using the DFT model, and those of amine-modified samples were evaluated using the BJH model. The main reason for this was that the DFT model could provide a reasonable and reliable approach for analyzing pore size distribution over the complete nanopore range [38]. Mesopore size is often assessed by the BJH method [39]. Their texture parameters are listed in Table 1. As shown in Figure 3a, the isotherms of $N_2$ adsorption–desorption for the HSMs(D) samples with dextran as a template exhibited a transitional sorption behavior between standard type I and IV, indicating the micro/mesoporous nature of the material. Meanwhile, the hysteresis loop in the $p/p_0$ range of 0.4–0.9 was assigned to the H4 type and further improved the existence of mesopores. The isotherms exhibited an increasing amount of adsorption at low relative pressure, suggesting the presence of micropores. In addition, the corresponding pore distributions for HSMs(D) samples are shown in Figure 3b; these samples exhibited several pore distribution peaks, mainly the presence of mesopores with a lightly broad distribution of pore diameters centered within the range of 2.0–4.0 nm, along with micropores less than 2.0 nm in size, which agreed well with the TEM-evaluated results. It is worthwhile to note that the HSMs(D)-30 sample possessed narrow micropores; the pore distribution was less than 1.0 nm as calculated by the DFT method. As shown in Table 1, the samples of HSMs(D) had middle BET surface areas in the range from 269 to 406 $m^2/g$ and pore volumes in the range of 0.13–0.23 $cm^3/g$. When the reaction temperature increased to 50 and 80 °C, compared with HSMs(D)-30, the micropore surface area for HSMs(D)-50 and HSMs(D)-80 decreased, whereas the mesopore surface area increased, suggesting that the reaction temperature was a key factor that affected the micro/mesoporous structure of HSMs(D).

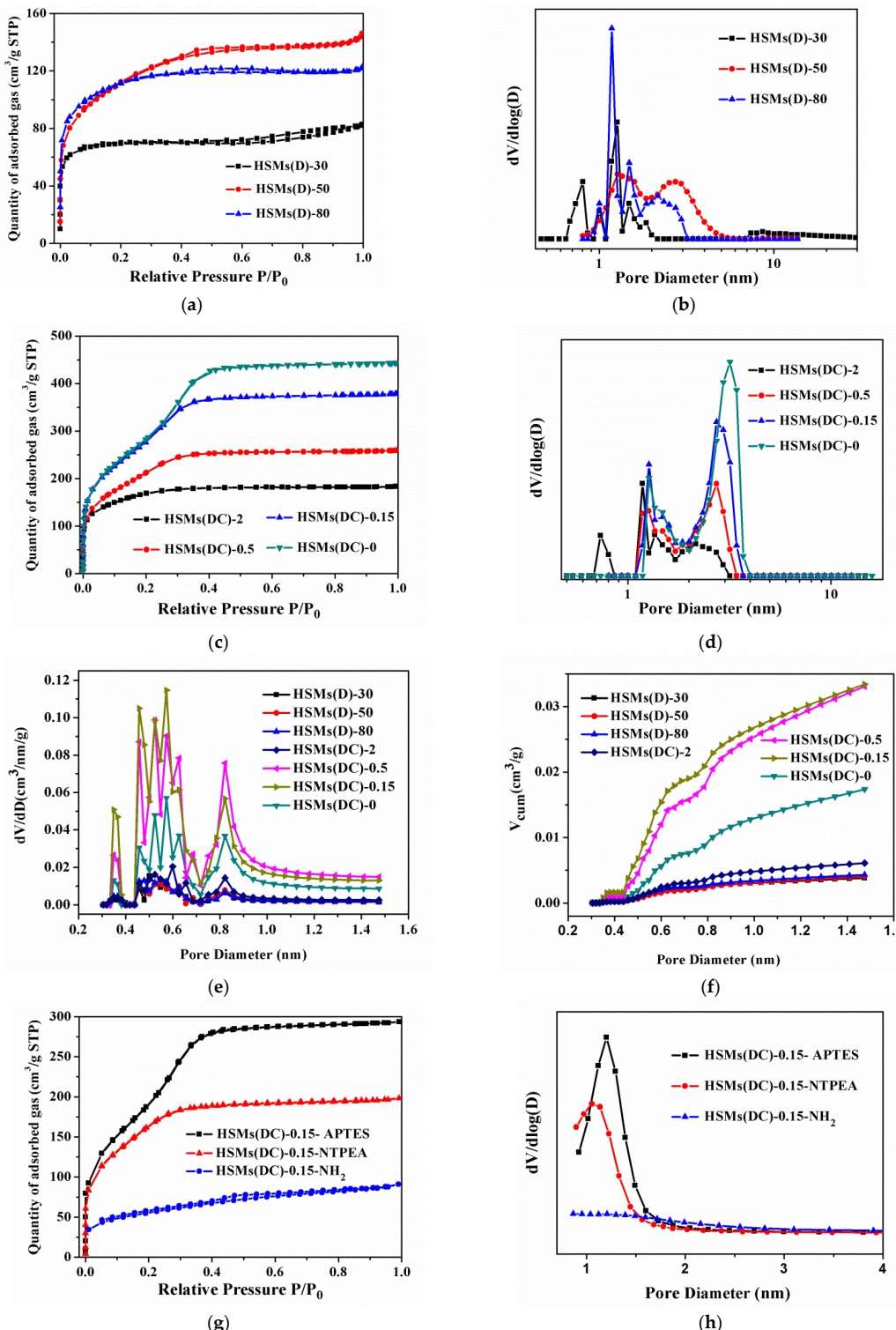

**Figure 3.** N$_2$ adsorption-desorption isotherm (**a**) for HSMs(D)-30, HSMs(D)-50, HSMs(D)-80 and the corresponding pore size distributions (**b**) (using N$_2$ adsorption isotherms with DFT model); N$_2$ adsorption-desorption isotherm (**c**) for HSMs(DC)-2, HSMs(DC)-0.5, HSMs(DC)-0.15, HSMs(DC)-0 and the corresponding pore size distributions (**d**) (using N$_2$ adsorption isotherms with DFT model); the narrow micropore of pore size distribution (**e**) and pore volume (**f**) for HSMs(D)-30, HSMs(D)-50, HSMs(D)-80, HSMs(DC)-2, HSMs(DC)-0.5, HSMs(DC)-0.15, HSMs(DC)-0 (using CO$_2$ adsorption isotherms at 273 K with DFT model); N$_2$ adsorption-desorption isotherm (**g**) for HSMs(DC)-0.15-APTES, HSMs(DC)-0.15-NTPEA, HSMs(DC)-0.15-NH$_2$ and the corresponding pore size distributions (**h**) (using N$_2$ adsorption isotherms with BJH model).

In addition, as shown in Figure 3c, the $N_2$ isotherm of the HSMs(DC) samples with dextran and CTAB as composite templates displayed type I adsorption with a steep increase at a low relative pressure $p/p_0$ of less than 0.1, and it presents the characteristic of micropores. Correspondingly, the surface area of HSMs(DC) samples was obviously enhanced with the increase in CTAB content. Specifically, the BET surface areas of HSMs(DC)-2, HSMs(DC)-0.5, HSMs(DC)-0.15, and HSMs(DC)-0 were comparably increased from 601, 801, and 1045 to 1052 $m^2/g$. The pore volumes of these samples were enhanced from 0.28, 0.42, and 0.58 to 0.68 $cm^3/g$. The micropore surface area and pore volume also increased. The micropore ranges of HSMs(DC) were observed in the pore size distribution (Figure 3d) in the range of 0.5–2 nm and the mesopore size range of 2.0–4.0 nm. The probable distribution of mesopore size increased from 2.0 to 4.0 nm with the increase in CTAB content. These observations confirmed that silica with a rich hierarchical porous structure was successfully synthesized. Moreover, it can be speculated that in the reaction system, some of the CTAB molecules formed micelle agglomerates, and other CTAB molecules with dextran molecules self-aggregated into a micelles-like agglomerate structure because of the existence of hydrogen bond interactions between CTAB molecules with amine groups and dextran molecules with hydroxyl groups. Similar results were reported by other researchers [40]. Therefore, this demonstrated that the hierarchical porous structure of the HSMs(DC) mainly originated from the composite template of the dextran/CTAB and CTAB content in the composite template, which played an important role in adjusting the pore structure of the micro/mesoporous silica.

It is well known that $N_2$ adsorption via the physisorption filling of wide micropores (D > 0.7 nm) occurs at very low pressures [41], while the high saturation vapor pressure (~3.5 MPa) of $CO_2$ at 273 K makes studying its adsorption an acceptable method for investigating materials with very narrow micropores (D = 0.3–1.0 nm) [42]. To further explore the narrow micropore structure (0.3–1.0 nm), the pore size distributions and pore volume of HSMs samples were investigated by analyzing $CO_2$ isotherms at 273 K and $p/p_0 < 0.03$ using the DFT method. The pore size distribution in Figure 3e indicates the HSMs samples with a broad pore size distribution, including super-micropores (D = 0.7–1.0 nm) and ultra-micropores (D = 0.3–0.7 nm). This phenomenon is uncommon in that these sorbents with narrow micropores were in favor of the uptake of $CO_2$. The related results of the narrow micropore volume from the DFT method are shown in Figure 3f and Table 1. The narrow micropore volume (range 0.003 to 0.034 $cm^3/g$) was much smaller than the $V_{micro}$ value (range 0.03 to 0.61 $cm^3/g$) by $N_2$ isotherms using the t-plot method because only the pore size in the ranged 0.3–1.0 nm was calculated by $CO_2$ isotherms. This result demonstrated that the proportion of narrow micropores in HSM samples was relatively low.

Furthermore, the isotherms of $N_2$ adsorption–desorption and pore size distribution profiles for HSMs(DC)-0.15 after grafting with amine groups are shown in Figure 3g,h. Compared with the unmodified HSMs(DC)-0.15 (Figure 3c), it was found that the surface area and pore volume of the modified samples decreased. As shown in Table 1, the surface areas for HSMs(DC)-0.15-APTES, HSMs(DC)-0.15-NTPEA, and HSMs(DC)-0.15-$NH_2$ descended to 781, 603, and 120 $m^2/g$ compared with those of HSMs(DC)-0.15. This change in the textural properties confirmed the successful grafting of the silica surface with the amine groups and was supported by the FTIR spectrum in the following section.

Figure 4 shows the FTIR spectra of HSMs(DC)-0.15 and HSMs(DC)-0.15-APTES. As can be seen in Figure 4, the spectrum of HSMs(DC)-0.15 exhibited adsorption peaks at 1085 and 803 $cm^{-1}$, attributed to the stretching vibration of Si-O-Si, and the adsorption band at 1628 $cm^{-1}$ was due to the bending vibration of adsorbed water molecules [43]. After modification, the new band at 3325 $cm^{-1}$ of HSMs(DC)-0.15-APTES was assigned to -NH asymmetric stretching vibration in the primary amine group, and the bands at 2937 $cm^{-1}$ and 2873 $cm^{-1}$ corresponded to the -CH asymmetric stretching and symmetric stretching vibration in the alky chain. The two weak bands at 1545 $cm^{-1}$ and 1475 $cm^{-1}$ were assigned to the symmetric bending vibration of -NH and bending vibration of C-

H [44,45]. The spectra of HSMs(DC)-0.15-NPTEA and HSMs(DC)-0.15-NH$_2$ are shown in Figure S3 (Supporting Information).

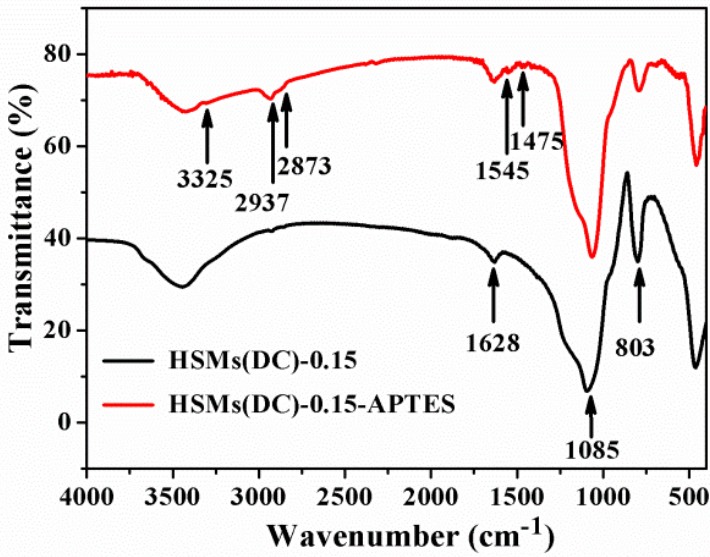

**Figure 4.** FTIR spectra of samples.

Figure 5 reveals the TGA profiles of HSMs(DC)-0.15, HSMs(DC)-0.15-APTES, HSMs(DC)-0.15-NTPEA, and HSMs(DC)-0.15-NH$_2$. As can be seen, it was observed that the weight loss of all samples before reaching 200 °C was attributed to the adsorbed water molecules and the residual solvent [46]. Furthermore, Figure 5 shows that the weight loss of approximately 2.92 wt% for HSMs(DC)-0.15 was recorded within the range of 200–800 °C, which generally corresponded to the dehydration of the silanol groups in the pore channels and residue of organic components. After amine modification with APTES and NTPEA, the weight loss of HSMs(DC)-0.15-APTES and HSMs(DC)-0.15-NTPEA reached 5.38 and 5.45 wt%, respectively, in the range of 200–800 °C, because of the decomposition of the organic functional group. Therefore, the amine-modified amounts were calculated to be ca. 2.46 and 2.53 wt%, respectively, after subtracting the weight loss of approximately 2.92 wt% for HSMs(DC)-0.15. Comparably, for HSMs(DC)-0.15-NH$_2$, as shown in Figure 5, the weight loss was enhanced, and a weight loss of approximately 19.23 wt% for HSMs(DC)-0.15-NH$_2$ was recorded within the range of 200–800 °C. The amine-modified HSMs(DC)-0.15 samples were determined by the weight loss observed in the temperature range, and the -NH$_x$ contents for the HSMs(DC)-0.15-APTES, HSMs(DC)-0.15-NTPEA, and HSMs(DC)-0.15-NH$_2$ were calculated to be 1.11 mmol/g, 1.14 mmol/g, and 86.8 mmol/g, respectively, and the maximum amine-loading capacity was obtained for HSMs(DC)-0.15-NH$_2$. This result indicated that APTES molecules and TEOS underwent condensation reactions and formed organic silica and silica framework, which provided a higher loading of the organic functionalities in the silica framework for HSMs(DC)-0.15-NH$_2$. Thus, considering the TGA results of the FTIR spectra, it could be concluded that amine groups were successfully modified on the surface and pore channels of HSMs(DC)-0.15.

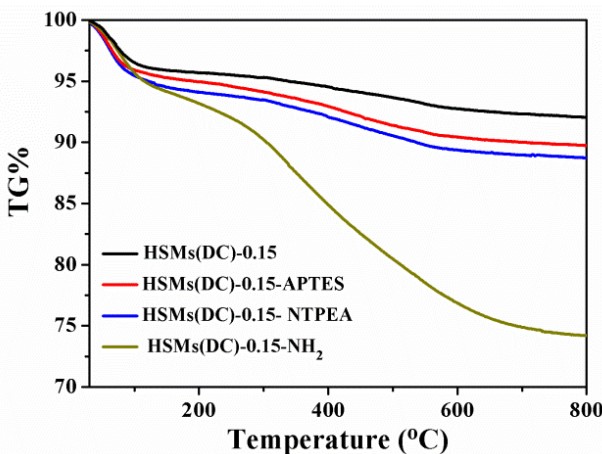

**Figure 5.** TGA curves of samples.

*3.2. CO$_2$ Adsorption*

Figure 6 shows the CO$_2$ adsorption isotherms of HSMs(D) and HSMs(DC), and Table 1 lists the CO$_2$ adsorption capacities measured at 273, 298 K, and 1.0 bar. As can be seen in Figure 6, the amount of CO$_2$ adsorption for the unmodified series of HSMs(D) and HSMs(DC) ranged from 1.00 to 1.64 mmol/g. All the isotherms exhibited low adsorption contents at low pressure and high adsorption contents at high pressure, suggesting that the isotherm adsorption profiles were attributable to the predominance of physical adsorption processes. Moreover, it was observed that these samples of CO$_2$ adsorption performance correlated with the pore structure. As shown in Figure 6, the order of adsorption capacity for HSMs(D) series at 273 K and 1.0 bar was HSMs(D)-80 > HSMs(D)-30 > HSMs(D)-50; the adsorption performance was not directly related to the specific surface area and pore volume. As shown in Table 1, although HSMs(D)-30 had the lowest BET-specific surface area and pore volume, its CO$_2$ adsorption capacity of 1.1 mmol/g was slightly more than that of HSMs(D)-50 (1.0 mmol/g) because of the increased microporous content of HSMs(D)-30 ($S_{mi}$ = 210 m$^2$/g) compared with that of HSMs(D)-50 ($S_{mi}$ = 70 m$^2$/g). Moreover, HSMs(D)-80 had the highest CO$_2$ adsorption capacity of 1.22 mmol/g due to the relatively high specific surface area and appropriate micropore content. As shown in Figure 6, the CO$_2$ adsorption capacity was in the following order: HSMs(DC)-0.15 > HSMs(DC)-0 > HSMs(DC)-0.5 > HSMs(DC)-2, and the highest CO$_2$ adsorption amount reached 1.64 mmol/g for HSMs(DC)-0.15 at 273 K and 1.0 bar. Comparability, for the HSMs(DC) series, an increase in CO$_2$ adsorption capability was observed. This result was mainly attributed to the higher micropore surface area and larger pore volume compared with those of HSMs(D). According to the previous discussion, high CTAB content in the composite template was beneficial for improving the micropore surface area and mesopore size of the hierarchical porous silica. Therefore, the effect of the pore structure of HSMs(DC) played an important role in the improvement of CO$_2$ adsorption capability. The samples with higher micropore surface areas and pore volumes exhibited an enhanced ability to capture CO$_2$ molecules. Furthermore, the existence of mesopores in the hierarchical porous silica material facilitated the flow of CO$_2$ into the material because of the smaller diameter of the CO$_2$ molecule (CO$_2$: 3.3 Å) compared with that of the mesopore [47]. In addition, as the adsorption temperature rose to 298 K, as shown in Figure S4 (Supporting Information), the CO$_2$ adsorption capability of the samples significantly declined, confirming that the interaction between the CO$_2$ molecule and sorbent surface was attributable to physical absorption.

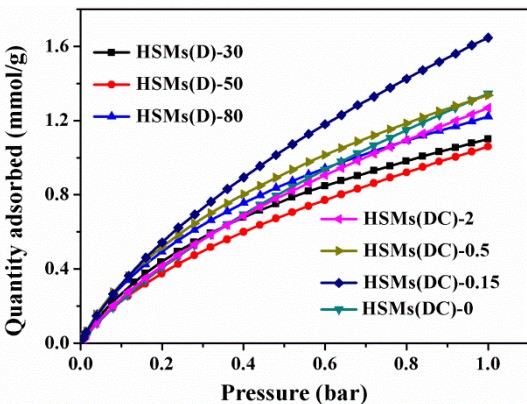

**Figure 6.** $CO_2$ adsorption isotherms of samples at 273 K.

Figure 7 displays the $CO_2$ adsorption isotherms of HSMs(DC)-0.15-APTES, HSMs(DC)-0.15-NTPEA, and HSMs(DC)-0.15-NH$_2$ modified with amine groups. The amounts of adsorbed $CO_2$ were 1.98 mmol/g for HSMs(DC)-0.15-APTES, 2.22 mmol/g for HSMs(DC)-0.15-NTPEA, and 1.38 mmol/g for HSMs(DC)-0.15-NH$_2$ at 273 K and 1 bar. It was found that the $CO_2$ adsorption capacity of HSMs(DC)-0.15 improved after modification with APTES and NTPEA through the post-grafting method. The surface area and pore volume of HSMs(DC)-0.15-APTES were higher than those of HSMs(DC)-0.15-NTPEA, while the amount of $CO_2$ adsorption HSMs-0.15-NTPEA was higher than that of HSMs(DC)-0.15-APTES. On the basis of the corresponding surface area and weight loss data, the obtained -NH$_x$ densities were $1.41 \times 10^{-3}$ mmol/m$^2$ for HSMs(DC)-0.15-APTES and $3.7 \times 10^{-3}$ mmol/m$^2$ for HSMs-0.15-NTPEA. Compared with HSMs(DC)-0.15-APTES, HSMs(DC)-0.15-NTPEA, with its higher -NH$_x$ density, had a higher amount of $CO_2$ adsorption [48]. These results suggest that the key parameters of the $CO_2$ adsorption capability of amine-modified materials are not only surface area and pore volume but also the density of -NH$_x$ on the surface of the matrix. Meanwhile, as shown in Figure 7, the capacity of $CO_2$ adsorption for HSMs(DC)-0.15-NH$_2$ declined to 1.38 mmol/g, which was lower than that of unmodified HSMs(DC)-0.15. As evidenced by N$_2$ adsorption–desorption measurement, the main reason was that the pore was filled by APTES molecules; therefore, a certain number of pores was blocked and inaccessible to $CO_2$ molecules, resulting in a low $CO_2$ sorption capacity.

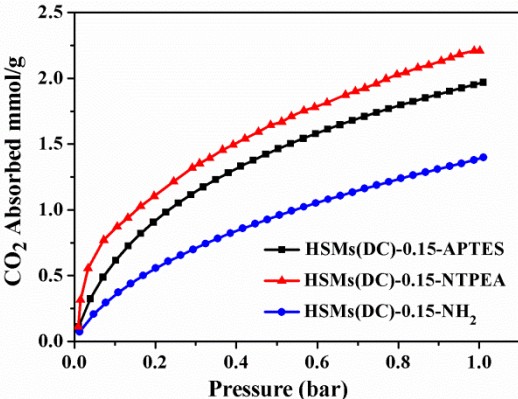

**Figure 7.** $CO_2$ adsorption isotherms of samples after amine modification at 273 K.

## 4. Conclusions

In conclusion, a series of hierarchical porous silica materials with abundant micropores and ordered mesopores were synthesized by the sol-gel method using dextran, dextran–CTAB, and CTAB as templates. In these HSMs, the pore structure of micro/mesoporous silica was dependent on the reaction temperature and the varying CTAB content With

an increase in the reaction temperature, the order degree of the mesopores for HSMs(D) samples with dextran as a template gradually decreased, and the specific surface area increased. The surface areas of HSMs(DC) samples with dextran and CTAB as templates were prominently enhanced with the increase in CTAB content in the composite due to the content of micropores in the hierarchical porous silica, and the mesopore size was improved. Additionally, to investigate the effect of hierarchical porous structure on the $CO_2$ capacity, HSMs were used as adsorbents to capture $CO_2$. All the samples exhibited an adsorption $CO_2$ capability; the detailed research demonstrated that the abundant microporous structure and large mesopore size were the keys to achieving improvement in $CO_2$ capacity. Furthermore, three modified samples with amine groups were tested as $CO_2$ capture adsorbents. The sorption capacity was dependent on the surface density of the amine groups and on the specific surface area of the hierarchical porous silica. In the future, the development of high-performance hierarchically porous silica materials with amine-functionalization for the capture of $CO_2$ is necessary, and it is expected to show feasibility for industrial applications.

**Supplementary Materials:** The following supporting information can be downloaded at: https://www.mdpi.com/article/10.3390/inorganics10070087/s1, Figure S1: XRD patterns of HSMs(D)-50 and HSMs(D)-80; Figure S2. SEM images of HSMs(DC)-0.15; Figure S3: FT-IR spectra of HSMs(DC)-0.15-NH2 and HSMs(DC)-0.15-NPTEA; Figure S4: $CO_2$ adsorption isotherms at 298 K of samples.

**Author Contributions:** X.J. synthesized and characterized the HSMs and carried the main responsibility for the writing of the manuscript. J.G. supervised the preparation work and contributed to writing the manuscript. L.Z. (Liyuan Zhang) carried out the adsorption of $CO_2$ experiment. Z.W. was responsible for analyzing the obtained data. L.Z. (Linlin Zhu) assisted X.J. in the review and editing of the original draft. M.X. supervised project administration. All authors have read and agreed to the published version of the manuscript.

**Funding:** This work was financially supported by the key research and development project of Anhui Province (Grant No. 202004a05020017); the Major Research Project of Natural Science from the Provincial Bureau of Education, Anhui, China (Grant No. KJ2019ZD62, No. KJ2021ZD0140); the Key Research Project of Natural Science from the Provincial Bureau of Education, Anhui, China, (Grant No. KJ2020A0748); the high-level scientific research and cultivation projects of Bengbu University (Grant No. 2021pyxm09); and the Bengbu University doctoral research funding project (BBXY2018KYQD22).

**Acknowledgments:** We extend our appreciation to the Anhui Province Engineering Laboratory of Silicon-based Materials for funding this work.

**Conflicts of Interest:** The authors declare no conflict of interest.

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
