# Peer review of "Synthesis of Hierarchically Ordered Porous Silica Materials for CO2 Capture: The Role of Pore Structure and Functionalized Amine"

_inorganics, doi:10.3390/inorganics10070087_

Round 1

Reviewer 1 Report

The manuscript, Synthesis of ordered hierarchically porous silica materials for 2 CO2 capture: the role of pore structure and functionalized  amine, should be revised and carefully checked. I suggest the following changes and corrections:

 In the Experimental part

Line 165-166   “The dextran template was extracted followed by the calcination at 550 oC for 6 h.”

How was dextran extracted, in which solvent, at what temperature, etc.?

In figure 3 it should be put the experimental points of the N2 adsorption-desorption isotherms and also in the pore size distribution curves. The authors must replace pore width with pore diameter and modify the title of the y axis with dV/dlogd (whithout pore volume because is wrong). In the isotherm graphs correct the title of y axis with Quantity of adsorbed gas.

For the discussion of the FTIR spectra the authors must correct the term of peak with band or vibration. Among mentioned bands from FTIR spectra there are also the deformation vibration of amine groups.

The authors should provide the amount (number) of amine groups on 1 g of silica, not only to describe the TGA analyses.

Generally, the English of the manuscript should be improved and checked because there are several mistakes of grammar.

The conclusions could be improved by inclusion some perspectives of the work. An optimisation of the silica matrix in term of amine groups and porosity is necessary.

 Minor corrections:

Line 31 ………….  Emissive emission carbon dioxides should be replaced with Emission of Carbon dioxide

Lines 62, 63 "This research demonstrated that silica supports with 3D pore structure exhibited higher  CO2 sorption capacity and amine efficiency than those with 2D-structured support." replace with: This research demonstrated that silica supports with 3D pore structure exhibited higher amine functionalization efficiency and thus, better CO2 sorption capacity than those with 2D-pore structure.

Line 119  - ... nature polymer dextran .... must be replaced with: natural polymer, dextran

Line 120 " …….. the effect of the reaction temperature and additive CTAB content ..." must be replaced with: the effect of the reaction temperature and CTAB content

Line 162 - "… this mixture solution was further added …….."

replace with:  ………… this mixture was further added ……..

Line 319  “…….. unmodified serials of HSMs(D) and  HSMs(DC)…” replace with ….. unmodified series of HSMs(D) ….

Author Response

Firstly, we are very appreciative to your suggestions and comments, which are very helpful for further improving quality of our manuscript. According to the reviewer’s comments and queries, the original manuscript has been revised. Meanwhile, we have reviewed the whole manuscript carefully and the modifications made in the revised manuscript have been highlighted in red color.

The detailed responses to reviewer’s comments are given in the response letter.

Reviewer 2 Report

Reviews for Inorganics: Synthesis of ordered hierarchically porous silica materials for CO2 capture: the role of pore structure and functionalized amine

The paper deals with the synthesis and characterization of hierarchically ordered porous silica for CO2 capture application. The synthesis of materials was done by sol-gel method with dextran, dextran/CTAB and CTAB as templates, and a surface modification was realized with amine groups. The effect of temperature of reaction during the synthesis was studied, as well as the influence of the amount of CTAB used or the amine functionalization, on the porous characteristics of the materials and on CO2 adsorption capacity. The paper is interesting, and a lot of work is presented for synthesis and characterization, however the paper lacks clarity, the nomenclature is sometimes confusing, the literature review should be actualized and English language should be improved. That is why I suggest accepting this paper after major modifications.

Comments:

-          The authors seem to make confusion between adsorption and absorption. The words absorbent or absorption are often used instead of adsorbent or adsorption. What is the phenomenon considered here? Authors should be more precise and use the right terminology.

-          At the end of the abstract, authors should remove « of the article and it must […] the main conclusions. »

-          Please remove « s » at « carbon dioxide »

-          English language should be improved.

-          P2, l55-59: the sentence is not clear. L59-63 is not clear: when authors say “silica supports with 3D pore structure exhibited higher CO2 sorption capacity and amine efficiency”, the higher CO2 sorption capacity was observed for modified with amine or unmodified silica?

-          P2 l80: when using “adsorbents and catalysis” do the authors mean “adsorbents and catalysts”?

-          P2 l82, please replace “hierachally” by “hierarchically”

-          P3 l 98-100: the sentence should be reformulated.

-          The title of paragraph 2.1 should be changed as only materials are described and not the methods in this part.

-          In part 2.2: how the samples were prepared for the TEM characterization? Were the samples dried before SEM/TEM characterizations?

-          P4 l151-154: this part referred to the N2 adsorption isotherm which is not indicated and should be placed l147 after the presentation of the surface area and porosity measurement.

-          P4 l176, Be careful with the chosen nomenclature, just before the samples are called HSM(DC)-0.15 and not HSM-0.15. Same comment for l184 authors speak about HSM-0.15-APTES but in the rest of the paper the sample is called HSM(DC)-0.15-APTES. Authors should more precise and consistent in their nomenclature.

-          P5 l201: where is Fig 1a? Where are the d(100) space values authors are referring in the text?

-          P7: the points a, b and c should be explained in the experimental part.

-          P7 l235: are authors sure type I isotherm is observed in Fig 3a?

-          P9 l317: HMD(DC) should be replaced by HSM(DC)

-          References: old references are used, only 6 papers on 44 are from 2018 or more. The literature review should be actualized with more recent papers.

Author Response

(The authors gave the same response as above.)

Reviewer 3 Report

To Authors

The revised manuscript deals with the efforts to the synthesis of porous silica materials with hierarchical micro/mesochannels structures, additionally superficial functionalized with amine moieties for the purposes of CO2 adsorptive capturing. The authors have undertaken the synthesis of these materials adjusting the structure-directing agents (dextran and CTAB). There was tested also the influence of the time of synthesis on the final structure of the solids. The resultant materials were characterized by means of low-temperature adsorption-desorption of nitrogen, low-angle XRD, FT-IR spectroscopy, SEM and TEM imaging, TG, and adsorption of carbon dioxide at 273 and 298 K. Although the paper is well-organized and the set of characterization methods was chosen properly, there are some issues to be addressed before the publication of this manuscript. These are the following:

1.            A thorough linguistic correction is required. There are some typos, certain sentences are difficult to understand. This makes the entire text unintelligible.

2.            The terms “adsorption” and “absorption” (see the discussion of textural parameters as well as FT-IR) are used interchangeably, although these are two different terms!

3.            The description of the pore size distribution computation models is incomprehensible: “the pore size distributions were calculated using adsorption branches of the isotherm by the Barrett-Joyner-Halenda (BJH) method and the adsorption branches of isotherms by density functional theory (DFT).” (L151-153). So, which model was employed in fact? Considering the curves displayed in Fig. 3, I suppose that for some samples the NLDFT model was used (see Fig. 3b and 3d), while for others the BJH approach was chosen (see Fig. 3f). There are no explanations neither in the figure caption nor in the discussion. What was the cause of employing two different calculation models? Please, address this point.

4.            Fig. 3e, for the sake of comparison, the isotherm of unmodified HSMs(DC)-0.15 sample should be provided in this panel.

5.            Table 1 and Fig. 3e, the authors provide the specific surface area for the sample HSMs(DC)-0.15-NH2 to be 120 m2/g, but the isotherm shows the nitrogen uptake close to nil. Thus, such a high surface area, in this case, is just impossible.

6.            Table 1, last column headline should be “Carbon dioxide uptake”.

7.            The term “ultramicropores”, according to the IUPAC recommendations, are those with a pore size of <0.7 nm, not 1.0 nm (as stated in L245).

8.            As the authors put efforts to perform the experiments of CO2 adsorption, it would be expedient to calculate the micropore size distributions from these CO2 isotherms (from the relative pressure region of p/p0<0.03). Also, the micropore volume might be calculated from these data.

9.            The authors state that: “Moreover, the rich hierarchical porous structure was obvious and directly observed from the TEM images, showed in Figure 2a-1, b-1 and c-1. The higher magnification TEM images (Figure 2 a-2, b-2 and c-2) demonstrated the existence of circle microporous, especially for HSMs(DC)-0.15 and HSMs(DC)-0 the worm-like mesoporous structure was found to co-exist in the same phase of micropores.” (L217-221). This conclusion seems to be speculative considering the SEM and TEM images.

10.          A general comment on the XRD patterns: although authors discuss the presence of the reflection at ca. 2 degrees two theta, they provide neither the Miller indices nor the space group of their materials. Such patterns resemble to some extent the XRD for MCM-41 hexagonal ordered silica (p6mm). Indeed, the MCM-41 is synthesized also with the CTAB surfactant. Please, address this issue.

In view of the above comments, I suggest the Editor refer the manuscript for major revisions.

Author Response

Title: Synthesis of ordered hierarchically porous silica materials for CO2 capture: the role of pore structure and functionalized amine

Corresponding author: Dr. Xiaoqi Jin 

Firstly, we are very appreciative to your suggestions and comments, which are very helpful for further improving quality of our manuscript. According to the reviewer’s comments and queries, the original manuscript has been revised. Meanwhile, we have reviewed the whole manuscript carefully and the modifications made in the revised manuscript have been highlighted in red color.

The detailed responses to reviewer’s comments are given in the response letter.

Round 2

Reviewer 1 Report

The manuscript can be published in this form.

Reviewer 2 Report

The authors took into account all the previous comments and the article is clearer now. It remains some minor english mistakes in the text but the paper should be accepted.

Reviewer 3 Report

As all the issues have been addressed, I suggest accepting this manuscript for publication.